# Hydrogen Bonding Drives Helical Chirality via 10-Membered Rings in Dipeptide Conjugates of Ferrocene-1,1′-Diamine

**DOI:** 10.3390/ijms232012233

**Published:** 2022-10-13

**Authors:** Monika Kovačević, Dora Markulin, Matea Zelenika, Marko Marjanović, Marija Lovrić, Denis Polančec, Marina Ivančić, Jasna Mrvčić, Krešimir Molčanov, Valentina Milašinović, Sunčica Roca, Ivan Kodrin, Lidija Barišić

**Affiliations:** 1Department of Chemistry and Biochemistry, Faculty of Food Technology and Biotechnology, University of Zagreb, 10000 Zagreb, Croatia; 2Central Laboratory, BICRO BIOCENTRE, Ltd., 10000 Zagreb, Croatia; 3CytomEx, Prisavlje 8, 10000 Zagreb, Croatia; 4Department of Food Engineering, Faculty of Food Technology and Biotechnology, University of Zagreb, 10000 Zagreb, Croatia; 5Division of Physical Chemistry, Ruđer Bošković Institute, 10000 Zagreb, Croatia; 6NMR Centre, Ruđer Bošković Institute, 10000 Zagreb, Croatia; 7Department of Chemistry, Faculty of Science, University of Zagreb, 10000 Zagreb, Croatia

**Keywords:** biological activity, chirality, conformational analysis, Density Functional Theory (DFT), ferrocene, hydrogen bonds, peptidomimetic, X-ray

## Abstract

Considering the enormous importance of protein turns as participants in various biological events, such as protein–protein interactions, great efforts have been made to develop their conformationally and proteolytically stable mimetics. Ferrocene-1,1′-diamine was previously shown to nucleate the stable turn structures in peptides prepared by conjugation with Ala (**III**) and Ala–Pro (**VI**). Here, we prepared the homochiral conjugates of ferrocene-1,1′-diamine with l-/d-Phe (**32/35**), l-/d-Val (**33/36**), and l-/d-Leu (**34/37**) to investigate (1) whether the organometallic template induces the turn structure upon conjugation with amino acids, and (2) whether the bulky or branched side chains of Phe, Val, and Leu affect hydrogen bonding. Detailed spectroscopic (IR, NMR, CD), X-ray, and DFT studies revealed the presence of two simultaneous 10-membered interstrand hydrogen bonds, i.e., two simultaneous β-turns in goal compounds. A preliminary biological evaluation of d-Leu conjugate **37** showed its modest potential to induce cell cycle arrest in the G0/G1 phase in the HeLa cell line but these results need further investigation.

## 1. Introduction

Protein–protein interactions (PPIs) play a key role in vital cellular processes and are the “holy grail” of modern life sciences and medicine [1,2]. Protein–protein binding affinity is mainly due to hot spots at protein interfaces, which consist of four to eight amino acids arranged as α-helices, β-sheets, or turns [3]. Since dysfunction of PPIs causes diseases, the identification of inhibitors capable of mimicking the hot spots along the PPI interface and disrupting their formation is of great therapeutic importance [4,5]. Peptidomimetic inhibitors, designed considering the hydrophobicity and complementarity of the surface residues of the PPI systems (less complementarity means weaker binding and easier disruption of PPIs in the presence of inhibitors), have been explored for the development of new drugs [1,6].

Peptidomimetics are defined as compounds whose essential pharmacophoric elements mimic a natural peptide or protein in 3D space to preserve their ability for interaction with the biological target and production of the desired biological effect [7]. Given their advantages over peptides and proteins, i.e., improved proteolytic stability, absorption, and selectivity, the concept of peptidomimetics as *the art of transforming peptides into drugs* represents a powerful tool in medicinal chemistry [8]. Moreover, the ability to design and synthesize peptidomimetics folded into predictable secondary structures contributes to their drug-like properties [9,10,11].

The most common secondary structural elements in proteins are β-turns (10-membered hydrogen bonded (HB) rings), the sites where the polypeptide chain folds to reverse its direction by almost 180° to form the globular shape. Their placement on the protein surface exposes them to cell receptors and also involves them in various biological interactions [12]. There are two main approaches to the design of β-turns: *the turn-inducing elements* approach, in which the amino acid at *i* + 1 and/or *i* + 2 is replaced by an element that can initiate the formation of the turn and restrict the conformational flexibility, and *the small molecular scaffolds as structural mimetics* approach, which is based on the replacement of the entire peptide backbone by the rigid scaffold that allows the alignment of the side chains in a spatial arrangement corresponding to the peptide turn residues [8].

The first approach was successfully applied to synthesize and conformationally evaluate a series of symmetrically disubstituted ferrocene peptidomimetics **I**–**III** consisting of turn-inducing scaffolds –OC–Fn–CO– (**I**) [13,14,15], –NH–Fn–CO– (**II**) [16,17], and –NH–Fn–NH– (**III**) [18,19]; (Fn = ferrocenylene), and amino acids (Figure 1). It was clearly shown that the distance of 3.3 Å between the cyclopentadienyl rings as well as the hydrogen bond donating potential of the scaffold used allowed the formation of intramolecular hydrogen bonds (IHBs) between strand, i.e., turn structures in ferrocene-amino acid conjugates **I** (two 10-membered IHB rings), **II** (9- and 11-membered IHB rings), and **III** (two 10-membered IHB rings) (Figure 1). The hydrogen bonds between the podand peptide strands in the bioconjugates **I**–**III** restrict the free rotation of the cyclopentadienyl rings and lead to helical chirality of the ferrocene moiety, as evidenced by strong Cotton effects near the absorption maximum of a ferrocene chromophore (around 470 nm) in CD spectra. The positive Cotton effects are related to the *P*-helicity of the ferrocene core, while *M*-conformers give rise to negative Cotton effects. Interestingly, a ferrocene chromophore also shows a potential to be used as a chiroptical probe for determining the helicity not only in disubstituted but in monosubstituted ferrocene derivatives as well. We have described a clear dependence between the sign of the CD spectra (near 470 nm) and torsion angle describing the inclination of the amide bond plane from the plane of the cyclopentadienyl ring [20,21,22,23,24].

The turn-inducing ferrocene scaffolds were then coupled with dipeptides to test the potential to form β-sheet-like hydrogen bonds between the strands in the corresponding higher homologues **IV**–**VI** (Figure 1) [25]. It was shown that the introduction of the dipeptides Ala–Pro, Ala–Phe, Gly–Leu, and Gly–Pro into the scaffold –OC–Fn–CO– induced helical chirality in the conjugates **IV** [25] and the introduction of Ala–Ala into –NH–Fn–CO– resulted in helically ordered conjugate **V**, as confirmed with Cotton effects in the ferrocene region [16]. The extraordinary CD activity (*M*_θ_ ~ 700,000 deg cm^2^ dmol^−1^) was observed when we introduced Ala–Pro sequences with alternating configuration into turn-inducing scaffold –NH–Fn–NH– to obtain dipeptide conjugates **VI**, and it was attributed to the formation of highly ordered turn structures [26]. It was shown that backbone chirality can be used to tune the preferred IHB ring size in peptidomimetics **VI**: the homochiral dipeptide conjugates were found to adopt the interstrand IHB patterns, realized through 10- and 13-membered IHB rings, while heterochiral peptides adopt the simultaneous intra- (7-membered) and interstrand (16-membered) IHB rings.

As found in the in vitro evaluation of the anticancer activity of the ferrocene peptidomimetics tested so far in our group [19,27,28,29], the lipophilicity of ferrocene conjugates **III** [19] was of crucial influence on the cytotoxic activity and compounds with larger retention factors (*R*_f_) showed better antiproliferative activity. The outstanding potential of the ferrocene scaffold –NH–Fn–NH– to induce a stable β-sheet-like structure in the conjugates **III** when coupled with Ala raises the question of whether its conjugates with other amino acids would adopt the same folding pattern and exhibit enhanced biological activity. With this in mind, our group has directed its research toward providing a library of homo- and heterochiral analogues of the compound **III** with a defined structure–activity relationship to serve as a simple model system for the selection of amino acids predisposed to contribute to the β-turn-mediated folding and biological potential of peptides **III**. 

Here, we report the synthesis, detailed conformational analysis, and biological evaluation of six homochiral enantiomeric conjugates **32**–**37**, obtained by replacing Ala from peptide **III** with amino acids having hydrophobic branched and bulky side chains: l- and d-Phe (**32**/**35**), l- and d-Val (**33**/**36**), and l- and d-Leu (**34**/**37**), respectively. 

We chose to study the homochiral enantiomeric peptides in this study for several reasons. It is well documented in the literature that homochiral amino acids, which adopt significantly fewer possible conformations compared to heterochiral amino acids, promote the formation of regular secondary structures [30]. Since the self-assembly of proteins is controlled by the chirality of the amino acid, their kinetic, morphological, and mechanical properties are significantly affected by the substitution of certain amino acids [31], i.e., the substitution of l- with a d-enantiomer or even a smaller part of the amino acid sequence can disrupt the helix or β-sheet, leading to destabilisation of the self-assembled peptides [32,33]. Since cell membranes are composed of chiral, enantiopure biomolecules, the altering of the peptide backbone chirality in terms of l- to d-substitution is expected to influence its biological features. In addition, Phe, Val, and Leu were recently reported to be involved in hydrogen bonding in β-hairpin mimetics [34,35], and we therefore expected their assistance in the formation of β-sheet-like structures in peptides **32–37**. However, considering the bulkiness of the side chains of Phe, Val, and Leu, which could interfere with hydrogen bonding, the formation of chirally ordered structures of different stability could be observed. We also expected that hydrophobic side chains of Phe, Val, and Leu would improve the overall hydrophobicity of the new peptides, which is a prerequisite for antibacterial and cytotoxic activity [36,37].

Due to its unique properties, i.e., air and thermal stability, superaromaticity, electrophilicity, solubility in organic solvents, easy functionalization, nontoxicity, and lipophilicity, ferrocene is widely used in medicinal chemistry for the derivatization of drugs and natural products [38]. Replacing the alkyl/aryl/heterocycle moiety of biologically relevant molecules with ferrocene or inserting ferrocene into an organic moiety/drug/drug-like molecule is considered a valuable tool for developing efficient and selective molecules to treat various diseases. In addition to anticancer activity related to its ability to generate ROS via oxidation of the central iron atom in the Fenton-like reaction, the ferrocene-derivatized scaffold exerts antioxidant, antituberculotic, antimalarial, anti-inflammatory, and antimicrobial activities. Moreover, compared to aryl or heteroaryl rings, the “barrel-shaped” ferrocene is better able to fill the hydrophobic cavity of the enzyme binding pocket, which contributes to its inhibitory potential. A comprehensive review of the above role of ferrocene in enhancing the bioactivity of organic scaffolds was recently given by Sharma and Kumar [39]. In addition, Ludwig et al. provided an overview of ferrocene conjugates with biomolecules used in a variety of infectious diseases caused by parasites, bacteria, fungi, and viruses [40]. The first report on the anticancer activity of ferrocenes, i.e., ferrocene polyamines, dates back to 1978 [41], and since then, ferrocene compounds have been investigated not only for their anticancer activity but also as potential antibacterial, antifungal, and antiparasitic drugs [42,43]. In a recent review of the anticancer activity of ferrocene conjugates with various biologically active molecules (amino acids/peptides, azoles, chalcones, coumarins, indoles, steroids, sugars, etc.), ferrocene–amino acid/peptide conjugates were highlighted as promising candidates for the development of new anticancer drugs due to their ability to overcome multidrug resistance during chemotherapy and to bind to specific receptors on cancer cells [44]. Metzler-Nolte and Albada reported on highly potent antibacterial ferrocene–peptide conjugates that are able to integrate into the bacterial membrane, resulting in detachment of vital enzymes needed for respiration and cell wall biosynthesis, making the bacteria more vulnerable [45]. The ferrocene group as a potent antioxidant has been used to modify and enhance the activity of natural antioxidants. The bioconjugates Fn–Orn–Orn–Orn and Fn–Tyr–Orn–Orn–Orn were evaluated as mimetics of the antioxidant enzymes superoxide dismutase and as inhibitors of peroxynitrite-mediated tyrosine nitration [46]. The introduction of ferrocene group into curcumin leads to higher antioxidant activity than the traditional hydroxyl-containing curcumin analogues [47]. In addition, ferrocene derivatives of caffeic acid and ferulic acid have shown the higher antioxidant properties compared to their phenolic parent acids.

Given the number of ferrocene conjugates tested to date for the treatment of infectious diseases, and the need to develop new bioactive ferrocene conjugates, we decided to test peptides **32**–**37** for their antitumor, antimicrobial, and antioxidant activities.

## 2. Results and Discussion

### 2.1. Synthesis of Peptides ***32***–***37***

The goal peptides **32**–**37** were synthesized according to the protocols established for the synthesis of ferrocene peptides **III** [19] and **VI** [26] (Figure 1). The *N*-terminus of Boc–NH–Fn–COOMe [48] was deprotected in the presence of gaseous HCl to afford the hydrochloride salt, which was treated with an excess of NEt_3_. The free amine obtained was then coupled with *C*-activated Boc–AA–OH (AA = l-Phe, l-Val, l-Leu, d-Phe, d-Val, d-Leu), respectively, to give carbamates **2**–**7** which were Boc-deprotected and converted to the acetamides **8**–**13**. The ester groups attached to the lower Cp rings of **8**–**13** were then carefully hydrolyzed to give the acids **14**–**19** (an equimolar amount of NaOH was used to prevent racemization, and the reactions were carried out at 80 °C for 1 h). The orthogonally protected goal precursors **26**–**31** were prepared by an in situ Curtius rearrangement of the unstable azides **20**–**25** obtained from acids **14**–**19**. To avoid the conversion of the intermediate isocyanate group to undesired *sym*-urea that could occur during the Curtius rearrangement, *t-*BuOH was freshly dried, and the reaction temperature was maintained at 65 °C. The conversion of azide (~2135 cm^−1^) to carbamate via isocyanate (~2270 cm^−1^) was monitored by IR spectroscopy and was complete when both IR bands disappeared. This was followed by acidic Boc-deprotection of the goal precursors **26**–**31** to give free *N*-termini, which were coupled with *C*-activated Boc–AA–OH to give the goal peptides **32**–**37**.

### 2.2. Computational Study

Systematic conformational analysis by means of computational chemistry methods can provide us with additional information on the intramolecular hydrogen bonding pattern and folding properties arising from a different side chain of the amino acids used in the synthesis. Over the past decade, we have used a hierarchical approach to study monosubstituted and disubstituted ferrocene-based derivatives [19,20,21,22,23,24,49,50,51]. The Monte Carlo Multiple Minimum and Mixed torsional/Low-mode sampling, as implemented in MacroModel [52,53], were used as a starting point for a relatively fast search of conformers with the OPLS2005 force field. A series of the most stable conformers was further optimized at the quantum mechanics level of theory (B3LYP-D3/LanL2DZ) and finally, with the B3LYP-D3 functional including the 6-311 + G(d,p) basis set (LanL2DZ for iron). Both optimizations were performed in a solvent modelled as a polarizable continuum. The presence of intramolecular hydrogen bonds was proven according to the criteria of Koch and Popelier [54] after Bader’s Quantum theory of atoms in molecules (QTAIM) analysis. More details can be found in the section DFT study and in the Appendix A).

The conformational analysis results were shown only for one pair of enantiomers of **32**–**34** (Ac–l-AA–NH–Fn–NH–l-AA–Boc; AA = l-Phe, l-Val, l-Leu). We expect the same Boltzmann distribution for each enantiomeric pair (**35**–**37**).

In all three compounds **32**–**34** (and in their enantiomeric pairs **35**–**37**), there is a relatively uniform distribution of conformers with a very robust motif consisting of two 10-membered rings connected by two hydrogen bonds between the substituents on the opposite cyclopentadienyl rings, NH_Fn_∙∙∙OC_Boc_ and NH_Fn_∙∙∙OC_Ac_, respectively, as shown in Figure 2. The other two NH groups (NH_Boc_ and NH_Ac_) are not involved in intramolecular hydrogen bonding, at least not in the most stable conformers. The pseudo-torsion angles defining the relative rotation of the two cyclopentadienyl rings (Figure 2a) remain smaller than 36°, thus defining the *P*-1,1′ isomer in the homochiral l-series and the *M*-1,5′ isomer in the d-series. Additional conformational flexibility arises from rotational freedom around the single C–C bond in the side chains of Phe, Val, and Leu, resulting in relatively small energy differences between the families of the most stable conformers with two 10-membered rings (two β-turns). Further details are provided in the Appendix A. These results can be compared with our previous study [19] on the very similar Ac–Ala–NH–Fn–NH–Ala–Boc system in which the alanine side-chain group, i.e., the methyl group, does not generate additional conformers by rotation around the single C–C bond. In the Ala derivatives, there was an energy gap between the most stable conformer (with two NH_Fn_ groups involved in two β-turns) and the other conformers in which three or all four NH groups were involved in hydrogen bonding. Very similar conformers were also found in the analysis presented here, but with higher relative energies, thus, with a very small contribution to the overall distribution of conformers. A special class of conformers in which there are no hydrogen bonds within or between strands is usually described as “open conformers” and can be used roughly to estimate the stabilizing effect of the hydrogen bonds. These structures are energetically less favorable (about 50 kJ mol^−1^, Appendix A) compared to the most stable conformer of each series confirming that two ten-membered rings are a thermodynamically very stable intramolecular hydrogen bonding motif.

### 2.3. IR Spectroscopy

It is expected that the stable conformations of **32****–37** predicted by the theoretical analysis will agree well with those determined experimentally. The experimental conformational analysis was performed for the l-enantiomers **32****–34**, and for their enantiomeric pairs **35****–37,** we expected and found identical scalar properties (Appendix A in the Appendix A), while for pseudoscalar properties such as the CD curves, we expected and found the same magnitude but opposite signs [55,56,57].

The experimental analysis of the conformations adopted by l–peptides **32–34** is based on spectroscopic techniques: concentration-dependent IR and NMR, temperature- and solvent-dependent NMR, NOESY NMR, and solvent-dependent CD.

Although both spectroscopic methods allow characterization of the specific part of the protein, the main advantage of IR over NMR is the ability to detect the majority of states, including those that fluctuate on fast time scales. If multiple distinct absorption bands belonging to the same vibration are observed, one can assume that multiple distinct environments or multiple conformations are present. However, the NMR time scale is slow, so signals from different states can rapidly interconvert and coalesce [58,59].

IR spectroscopy of backbone vibrations provides information about their hydrogen bonding engagement. When amide N–H or C=O groups are involved in HBs, their IR stretching frequencies are red shifted [60,61,62,63]. The amide A region (3300–3500 cm^−1^) of the studied peptides contains two distinct set of NH signals above and below 3400 cm^−1^, which is certainly due to the adoption of non-bonded and hydrogen-bonded conformations. The ratio of free and associated NH bands shows that the bulky benzyl side chains strongly interfere with hydrogen bonding (Figure 3a). A higher ratio of free and associated NH bands (0.7:1) indicates a higher proportion of non-bonded states in l-Phe conjugate **32** compared to conjugates with l-Val **33** and l-Leu **34** (0.5:1), which are therefore expected to adopt hydrogen-bonded conformations to a higher extent. This trend is confirmed by the vibrations of the carbonyl functions of the urethane and acetamide groups. The red shifting of hydrogen-bond-accepting urethane [64,65] and acetamide [66] groups was observed in our previous study of the Ala-conjugates **III** (below 1680 cm^−1^), while the C=O groups of the non-bonded reference compounds Fn–NH–Boc and Fn–NH–Ac were seen at 1723 cm^−1^ and 1684 cm^−1^, respectively [19] (Appendix A in the Appendix A). The hydrogen-bond-acceptor role of CO_Boc_/_Ac_ of the peptides studied here is strongly supported by sharp and narrow signals below 1680 cm^−1^. However, the blue-shifted shoulders of the non-bonded carbonyl functions are more pronounced in l-Phe conjugate **32** (Figure 3b). Considering that the intramolecular hydrogen bonds (IHBs) are required for the adoption of the expected β-turn structures, we tested the concentration dependence of the IR pattern. The ratios of the free and associated NH bands did not change when diluted from 5 × 10^−2^ M to 1.25 × 10^−2^ M, indicating the complete dominance of the IHBs (Appendix A in the Appendix A). Otherwise, the intensities of the intermolecularly engaged NH bands would have been reduced upon dilution compared with the free NH bands. This is in full agreement with the results of the DFT study and the dominance of conformers containing two NH_Fn_ groups involved in two 10-membered hydrogen-bonded rings and two free NH groups. In addition, the solid-state IR spectra of peptides **32**–**34** were characterized by a total prevalence of associated NH groups (Figure 3c and Appendix A in the Appendix A).

### 2.4. NMR Spectroscopy

NMR spectroscopic analysis not only reveals different magnetic resonances of the nuclei that depend on their chemical environment, but also allows the structural study of small peptides that tend to undergo conformational changes, resulting in an ensemble of conformers [67,68]. In this study, we performed detailed 1D (^1^H, ^13^C) and 2D NMR studies (^1^H-^1^H COSY, ^1^H-^1^H NOESY, ^1^H-^13^C HMQC, and ^1^H-^13^C HMBC) to assign the individual proton resonances and to determine the hydrogen-bonding patterns for the goal compounds. (For the full NMR characterization, see Appendix A). 

When the amide protons are accessible to the hydrogen bond acceptor, their resonances are moved downfield (δ ≥ 7 ppm in nonpolar CDCl_3_) [69,70]. In addition, stronger hydrogen bonds cause greater deshielding and thus, larger chemical shifts of the amide protons [71]. Therefore, the resonances observed here above 9 ppm for NH protons belonging to the ferrocene scaffold of peptides **32–34** and coinciding with the hydrogen-bonded NH_Fn_ in the Ala-analogue **III** [19] suggest their involvement in strong hydrogen bonds. The resonance signals of NH_Ac_ (δ~6.8–7.3 ppm) indicate a lower potential for strong hydrogen bonding, whereas the significantly upfield shifted NH_Boc_ (δ~5 ppm) are not expected to be involved in hydrogen bonds (Appendix A in the Appendix A) and therefore, their dependence on concentration, temperature, and polar solvent is not considered further.

The full IHB engagement of the associated amide groups, indicated by the IR data, was further tested by measuring the concentration dependence of the NMR chemical shifts (Appendix A in the Appendix A). Since the significantly downfield-shifted NH_Fn_ showed negligible upfield shifts (Δδ < 0.16 ppm) at high (50 mM) vs. low concentrations (6.25 mM), their involvement in IHBs is further supported. Although the concentration-independent IR spectra rule out the presence of intermolecular hydrogen bonds, the significantly altered chemical shifts of the acetamide NH_Ac_ protons at dilution (Δδ~0.6–1 ppm) nevertheless indicate the possible presence of intermolecular aggregates (Figure 4).

The chemical shift of the amide proton involved in hydrogen bonding depends on temperature. At increasing temperatures, the hydrogen bonding is weakened, and the chemical shift of the amide proton is increased, i.e., moved to the upper field [72]. This effect is less pronounced for amide protons involved in stronger IHBs and more stable conformations [73]. The signals of NH_Fn_ showed smaller upfield shifts (Δδ < 0.6 ppm), confirming the assumption that they participate in strong IHBs. Larger upfield shifts (Δδ > 1.2 ppm), experienced by NH_Ac_ from l-Phe-dipeptide **32** and l-Leu-dipeptide **34**, are certainly due to their involvement in less stable structures (Figure 5). Moreover, the observed peaks of hydrogen-bonded NH_Fn_ became sharper at lower temperatures, which may be attributed to the common slowing motion of the molecules [74], (see Appendix A in the Appendix A). Additional information about the hydrogen bonding of the amide protons was obtained by measuring the temperature coefficients (Δδ/Δ*T*), i.e., the changes in chemical shifts as a function of temperature. The low Δδ/Δ*T* values (−2.4 ± 0.5 ppb K^−1^) can be ascribed to initially shielded protons and protons exposed to the solvent (CDCl_3_). A larger temperature dependence corresponds to initially shielded protons (involved in hydrogen bonds), exposed to the solvent during the unfolding of the IHB-stabilized structures or dissociation of aggregates at higher temperatures [19,26,75,76,77,78]. The large temperature coefficients (−5.71 to −8.71 ppb K^−1^) observed here for concentration-independent NH_Fn_ reflect their involvement in IHB-mediated folding, whereas the possible engagement of NH_Ac_ in self-assembly resulted in large temperature coefficients (−11.57 to −18.42 ppb K^−1^). However, NH_Boc_ shows no significant dependence on temperature as it is not involved in HBs (Appendix A in the Appendix A).

Protein folding is controlled by a handful of noncovalent interactions that include the hydrophobic effect, conventional hydrogen bonding, Coulomb interactions, and van der Waals interactions [79]. Since IHBs are believed to be an important factor in the folding of the peptidomimetics tested, we attempted to determine their strength by titration with DMSO. Because of its strong hydrogen-bond-accepting ability, DMSO solvates exposed NH protons and their resonances are moved downfield. However, the amide NH protons involved in strong HBs are shielded from the solvent and their chemical shifts do not change significantly during titration with DMSO. As shown in Appendix A and Appendix A in the Appendix A, the chemical shifts of concentration- and temperature-independent NH_Fn_ were not significantly affected (Δδ ~ 0–0.16 ppm) in the presence of 55% DMSO (Figure 6), which is certainly due to their participation in strong IHBs. As expected, NH_Ac_, whose chemical shifts depend on concentration and temperature, showed considerable solvent sensitivity (Δδ ≥ 1–1.7 ppm), which may be attributed to their involvement in weaker HBs (Figure 6 and Appendix A in the Appendix A).

The model Ala-conjugate **III** was stabilised by two simultaneous 10-membered interstrand hydrogen bonds NH_Fn_∙∙∙OC_Ac_ and NH_Fn_∙∙∙OC_Boc_, i.e., by two simultaneous β-turns (Figure 1) [19]. To further investigate whether this conformational pattern is maintained upon replacement of Ala with Phe, Val, and Leu in peptides **32–34**, we analyzed the possible interstrand NOE interactions between NH_Fn_ and the *N*-terminal Boc or Ac group (Figure 7). Since the targeted interstrand NOE contacts between NH_Fn_ and the carbamate group were observed in peptides **33** and **34**, the presence of interstrand IHB NH_Fn_∙∙∙OC_Boc_ corresponding to β-turn is strongly supported. The lack of required NOE contacts for l-Phe peptide **32**, whose 1D NMR data indicate somewhat increased sensitivity to changes in concentration, temperature, and solvent polarity, is further evidence for its involvement in weaker IHBs.

### 2.5. CD Spectroscopy

Circular dichroism (CD) allows monitoring of the conformational changes that occur during aggregation and thermal and chemical unfolding of peptides and proteins [80,81]. When ferrocene peptides are folded into turn- and β-sheet-like conformations stabilized by IHBs, the free rotation of the ferrocene rings is restricted, resulting in helical chirality of the ferrocene core, which is reflected in the Cotton effects around 470 nm. The intensity of the Cotton effects is proportional to the stability of the folded conformation, and their positive or negative sign is due to the right- or left-handed helicity of the ferrocene core [82]. 

The presence of two simultaneous β-turns in the model Ala-conjugate **III** resulted in helical chirality of ferrocene, as evidenced by strong positive Cotton effects (*M*_θ_ ~ 23,000 deg cm^2^ dmol^−1^). Its CD-activity was not significantly decreased in the presence of DMSO (<30%), which was further evidence of its conformational stability [19].

Inspection of the CD spectra of peptides **32**–**37** reveals that (*i*) the enantiomeric pairs exhibit Cotton effects of opposite sign and the same (or very similar) intensity, and (*ii*) the bulking and brunching of the amino acid side chain affect the stability of the folded conformations. When comparing with model Ala-peptide **III**, replacement of the methyl side chains with bulky benzyl groups in **32/35** resulted in a negligible loss of CD activity (*M*_θ_~20,000 deg cm^2^ dmol^−1^), while branched isopropyl and isobutyl side chains in **33/36** and **34/37** improve their conformational stability, thus resulting in increased intensity of Cotton effects (*M*_θ_~27,000 deg cm^2^ dmol^−1^). To further confirm that the stable conformations of the studied peptides are realized via strong IHBs involving NH_Fn_ whose chemical shifts were not significantly affected by the addition of DMSO, we tested their CD behavior in the presence of 20% DMSO. CD data showed the preservation of 75–90% of DMSO-free activity, confirming that the competing solvent did not significantly perturb the obtained stable folded structures. The observed changes in hydrogen-bonding behavior are believed to occur in the water environment, which is due to the ability of water to accept and donate hydrogen bonds. The helical chirality of ferrocene was also noticed in the solid-state CD spectra of peptides **32**–**37**. As seen in the solution state, the enantiomeric pairs **32/35**, **33/36** and **34/37** exhibited the Cotton effects of the same intensity, but of the opposite sign (Figure 8d).

### 2.6. X-ray Crystal Structure Analysis

Although we put a lot of effort into obtaining single crystals for X-ray crystal structure analysis, only compounds **33** and **36** gave single crystals of suitable quality. The l-Val peptide **33** and its enantiomer d-Val peptide **36** adopt an intramolecular hydrogen bonding pattern predicted by computational study, i.e., two 10-membered rings (β-turns) (Figure 9 and Appendix A in the Appendix A), additionally stabilized by six intramolecular C-H···O hydrogen bonds (Table 1). Two cyclopentadienyl rings are in a staggered arrangement (rotated by approximately 39°), so the torsion angle between C1–N1 and C6–N3 bonds is 39.25°. The molecule comprises four NH (strong proton donors) and four carbonyl groups (strong proton acceptors). Two pairs of donors and acceptors (N1–H1···O5 and N3–H3···O2) are pointed ‘inwards’ and participate in intramolecular hydrogen bonding. The other two pairs (N2–H2···O4 and N4–H4···O1) are pointed ‘outwards’ towards neighboring molecules (symmetry operators 1/2 + *x*, 1/2–*y*, –*z* and −1/2 + *x*, 1/2−*y*, −*z*), and form two pairs of intermolecular hydrogen bonds (Table 1, Figure 9). In this way, infinite hydrogen bonded chains are formed in the direction [100] (Figure 10). In the direction [001], the molecules are held by C-H···π interactions between methyl groups and cyclopentadienyl rings, and in the direction [010], there are only dispersion interactions.

### 2.7. Biological Evaluation

After synthesis and conformational analysis, the prepared compounds were studied biologically in vitro. The antitumor activity of the peptidomimetics **32**–**37** was assessed by MTT test on a panel of three tumor cell lines of different origin, with additional analysis of apoptosis and cell cycle for peptidomimetic **37**. Antimicrobial activity of the goal compounds was studied against Gram-positive and Gram-negative bacteria, lactic acid bacteria, and yeasts using a disk diffusion method. Their antioxidant activity was evaluated by 1,1-diphenyl-2-picryl-hydrazyl radical scavenging assay and ferric reducing antioxidant power method.

#### 2.7.1. Antitumor Activity

IHBs are of crucial importance in biochemistry and chemistry. In addition to electronic distribution, IHBs affect the molecular geometry, shape, and conformation of bioactive molecules, thus revealing the significant impact on molecular properties, function, and interaction [83]. Compared with the model Ala-conjugate **III** [19], novel conjugates **32**–**37** adopt the similar IHB pattern, i.e., two simultaneous β-turns. Moreover, the IHB structures in Val- and Leu-conjugates **33/36** and **34/37** are slightly more stable than those in the Ala-peptide **III**, whereas the replacement of Ala by Phe in **32/35** causes a slight destabilization of the IHB structures. Considering the conformational similarity between the model peptide **III** (which was shown to exhibit antiproliferative activity against HepG2 cell line) and the novel peptides, we made a preliminary evaluation of their antiproliferative capacity.

The antitumor activity of compounds **32**–**37** was evaluated in vitro according to the NIH-NCI DTP protocol [84] on 3 different tumor cell lines: HeLa (cervical adenocarcinoma), HepG2 (hepatocellular carcinoma), and MCF-7 (breast adenocarcinoma). The study showed that the overall antitumor activity of the peptidomimetics studied was weak. On HepG2 cells, there was almost no inhibitory effect within the tested range, whereas the inhibition of cell growth was strongest on MCF-7 cells (Figure 11, Table 2). Interestingly, the strongest effect was detected in HeLa and MCF-7 cells with a different pair of peptidomimetics, Leu and Phe conjugates, respectively. Whereas l- and d-Leu peptides **34**/**37** showed inhibitory effect greater than 50% in HeLa cells, l- and d-Phe peptides **32**/**35** showed an inhibitory effect of more than 50% in MCF-7 cells, within the tested range. However, l- and d-Val peptides **33**/**36** had very little effect on all cell lines tested. Very similar results for the enantiomeric pairs **32/35**, **33/36,** and **34/37**, respectively, suggest that the type of side chain is more important than its absolute configuration for the inhibitory effect. In addition, very different cell lines (HeLa and MCF-7) were affected by different types of peptidomimetics (Leu-containing compounds **34/37** and Phe-containing compounds **32/35**, respectively), possibly indicating different modes of action, as these very different cell lines bear different weakpoints.

Based on our previous results on the antitumor activity of ferrocene peptidomimetics, it was found that lipophilicity contributes significantly to the biological activity, i.e., ferrocene peptidomimetics with larger retention factors (*R*_f_) in less polar solvents showed better antiproliferative capacity [19,27,28,29]. However, the results obtained here are only partially consistent with the previous findings and are more dependent on the cell line treated (the HepG2 cell line was resistant to the peptides tested, whereas the HeLa cell line was sensitive only to Leu-peptides **34/37**, and Phe-peptides **32/35** affected only the MCF-7 cell line). Compared with the more polar Val-conjugates **33/36** (*R*_f_~0.58) and Leu-peptides **34/37** (*R*_f_~0.45), Phe-conjugates **32/35** (*R*_f_~0.83) showed the highest cytotoxic effect with an IC_50_ values of 53.1 ± 23 μM and 32.7 ± 6.89 μM, respectively, as expected (Table 2). Compared with cisplatin, which is commonly used as a reference drug in antitumor assays, the obtained IC_50_ values for Phe-peptides **32** and **35** are even lower, whereas the IC_50_ value for d-Leu conjugate **37** is~2-fold higher (Table 2) [85].

In this work, we have explored whether apoptosis could be induced in HeLa cells, our in vitro model, by the compound **37**. Cells were treated with four different concentrations of the tested compound and analyzed 24 h post treatment by flow cytometry using the standard Annexin V-FITC/PI double-labeling method to detect viable, total apoptotic, and necrotic cells. In other studies, it was demonstrated that peptidomimetics elicited a pro-apoptotic effect [87]. According to our findings, there were no significant differences between the control and treated cells (Figure 12). However, we observed a higher percentage of total apoptotic cells in treated samples with the effect increasing along with the concentration of the tested compound. These results indicate the potential of this peptidomimetic in the induction of apoptosis, although further studies will be necessary to help consolidate our findings and to investigate the role of compound **37** on apoptosis in various cell line types.

To investigate the effect of peptide **37** on the HeLa cell cycle progression, we treated cells with different concentrations of the compound and analyzed the untreated and treated cells 24 h after treatment by flow cytometry (Figure 13). Our data showed that cells treated with 41 µM and 105 µM compound **37** exhibited a significant increase in the number of cells in the G0/G1 phase compared with control cells, corresponding to a significant decrease in the proportion of cells in the S phase (* *p* < 0.05). These results suggest that one of the mechanisms by which compound **37** may inhibit proliferation of cancer cells is the induction of cell cycle arrest in the G0/G1 phase, which is accompanied by a decrease in the percentage of cells in the S phase. In mammalian cells, cell cycle arrest in the G1 phase is one of the cellular responses induced when environmental conditions are unfavorable for cell division, such as the presence of DNA damage. This event prevents cells with damaged DNA from entering the S phase and allows them to repair their DNA. When DNA repair is not possible, the cells undergo apoptosis. A similar result was observed in cells treated with a concentration of 26 µM, where we observed more cells in the G0/G1 phase. On the other hand, treatment with a concentration of 66 µM significantly decreased the percentage of cells in the G2/M phase (* *p* < 0.05). These results, together with those shown in Figure 12, may indicate that compound **37** causes early apoptosis event as a results of cell cycle arresting. Because cell cycle regulation in mammalian cells is very complex, further studies should be performed to determine the exact mechanism underlying these effects.

In this work, we investigated the effect of d-Leu conjugate **37** on apoptosis and the cell cycle of HeLa cells. In our future studies, we will focus on testing the effect of homochiral Phe-conjugates **32** and **35** as well as the effect of their heterochiral analogues on apoptosis and cell cycle under the same experimental conditions as in this study. In addition, we believe it would be useful to test these compounds on other continuous cell lines of different origin.

#### 2.7.2. Antimicrobial Activity

The possible antimicrobial activity of the ferrocene peptides **32**–**37** was tested by the disk diffusion method. Unfortunately, the absence of a growth inhibition zone around the diagnostic disks indicates the resistance of the tested bacteria and yeasts to the ferrocene peptides **32**–**37** (Appendix A in the Appendix A). Therefore, the tested peptides showed no antimicrobial activity despite the high concentrations applied (1 mg of the compound was applied to the disk, ultimately corresponding to concentrations of 153 mM, 180 mM, and 171 mM for the ferrocene conjugates with Phe (**32** and **35**), Val (**33** and **36**), and Leu (**34** and **37**), respectively). However, the low minimal inhibitory concentrations (MICs) for ferrocene peptides with antimicrobial activity have been reported in the literature [45,88,89]. The pentapeptide composed of ferrocene and alternating arginine and tryptophan units exhibits even better activity against *S. aureus* (7.1 µM) than the toxin pilosulin 2, which was used as a positive control [88]. In addition, the MIC values of its antibacterial activity against *E. coli* (28–57 µM) and *MRSA* (28 µM) are much significantly lower than the values obtained in our work [45,89].

Since the antimicrobial activity of the goal compounds was not observed in the present study, we have tested in a preliminary way the inhibitory potential of the compounds **11**–**13** as precursors of the d-peptides **35**–**37**. The encouraging preliminary results clearly indicate the needs for further detailed studies aiming to provide the potent ferrocene antimicrobial agents that can combat increasing antibiotic resistance.

#### 2.7.3. Antioxidant Activity

While the antioxidant activity of the bioactive peptides has been described in detail and the relationship between their structural features and antioxidant activity has also been given [90], the antioxidant activity of ferrocene peptidomimetics has not been reported yet. Therefore, we decided to examine the possible antioxidant activity of peptides **32**–**37**. All the tested compounds showed moderate antioxidant activity in the range of 0.1 mM Trolox equivalent, evaluated by the DPPH method, and in the range of 0.52–0.63 mM Trolox, evaluated by the FRAP method (Table 3).

To the best of our knowledge, the above data are the first results on the antioxidant activity of this type of compounds. The antioxidant activity did not differ significantly among the tested compounds, although literature studies suggest that the antioxidant activity in biologically active peptides depends on the type and sequence of amino acids (a high proportion of hydrophobic amino acids was found in peptides with high antioxidant activity).

## 3. Materials and Methods

### 3.1. General Procedure and Methods

Peptides **2–37** were synthesized under argon atmosphere using chemicals of analytical purity. (The synthetic procedures and spectroscopic characterization of precursor **2**–**31** can be found in the Appendix A.) CH_2_Cl_2_ used for synthesis, CD measurements, and FTIR was dried (P_2_O_5_), distilled over CaH_2_, and stored over molecular sieves (4 Å). EDC (Acros Organics), HOBt (Aldrich), and acetyl chloride (Aldrich) were used as received. The synthesis of Boc–NH–Fn–COOMe **(1)** has been reported previously by us [41]. Its *N*-terminus was deprotected by action of gaseous HCl. The *N*-termini of l- and d-AA (AA = Phe, Val and Leu) were protected in the presence of sodium hydroxide, aqueous dioxane, and di-*tert*-butyldicarbonate to give Boc-l-AA-OH and Boc-d-AA-OH, respectively. Boc-l-AA-OH and Boc-d-AA-OH were activated with the coupling reagent HOBt for 1h in CH_2_Cl_2_. The products were purified by preparative thin layer chromatography on silica gel (Merck, Kieselgel 60 HF_254_) using EtOAc/CH_2_Cl_2_ mixture or pure EtOAc as eluent. Infrared spectra were recorded as CH_2_Cl_2_ solutions between NaCl windows or in KBr using a Bomem MB 100 mid FTIR spectrometer ((s) = strong, (m) = medium, (w) = weak, (br) = broad, (sh) = shoulder). The ^1^H and ^13^C NMR spectra were recorded at 600 MHz using the Bruker Avance spectrometer with a 5 mm TBI probe at Ruđer Bošković Institute, and were referenced to the peak of the residual solvent (CDCl_3_-*d*, ^1^H: *δ* = 7.24 ppm, ^13^C: *δ* = 77.23 ppm). In the case of the CDCl_3_-*d*/ DMSO-*d*_6_ mixture, calibration was performed using Me_4_Si as an internal standard (^1^H: *δ* =0.0 ppm). Double resonance experiments (COSY, NOESY, HMQC, and HMBC) were performed to facilitate the assignment of signals ((s) = singlet, (d) = doublet, (t) = triplet, (q) = quartet, (m) = multiplet, (dd) = doublet of doublets). Unless otherwise stated, all spectra were recorded at 298 K. NMR titrations were performed by adding 10 μL portions of DMSO-*d_6_* to NMR tubes containing CDCl_3_-*d* solutions of the peptides under study (*c* = 2.5×10^−2^ M). Spectra were recorded after each addition, and DMSO-*d_6_* was added until no change in the chemical shift of the amide protons was observed. CD spectra were recorded using a Jasco-810 spectropolarimeter in CH_2_Cl_2_ or KBr. Molar ellipticity coefficients (*θ*) are given in degrees, concentration *c* in molL^−1^, and path length *l* in cm, so that the unit for (*θ*) is deg cm^2^ dmol^−1^. Mass spectra were recorded using HPLC-MS system coupled to a triple–quadrupole mass spectrometer, operating in a negative ESI mode. High-resolution mass spectra were recorded using a 4800 MALDI TOF/TOF-MS analyser (see Appendix A). Melting points were determined using Reichert Thermovar apparatus. Single crystal measurements were performed with an Oxford Diffraction Xcalibur Nova R.

#### 3.1.1. Synthesis of Ac–l-AA–NH–Fn–NH–AA–Boc (**32**–**37**)

The HCl_gas_ was purged through the suspension of Ac–l-AA–NH–Fn–NH–Boc (**26**–**28**) and Ac–d-AA–NH–Fn–NH–Boc (**29**–**31**) (1 mmol) in dry CH_2_Cl_2_ (5 mL) at 0 °C, respectively. After approx. 30–120 min, the solvent was evaporated in vacuo, and the obtained hydrochloride salt was then suspended in CH_2_Cl_2_ and treated with NEt_3_ (pH~8) to give an unstable free amine suitable for coupling to Boc-l-AA-OH or Boc-d-AA-OH (2 mmol) using the standard EDC/HOBt method (EDC (3 mmol); HOBt (3 mmol)). The reaction mixtures were then stirred at room temperature until the ferrocene amine was completely consumed, which was monitored by TLC (~2 h). Standard work-up (washing with a saturated aqueous solution of NaHCO_3_, a 10% aqueous solution of citric acid and brine, drying over Na_2_SO_4_ and evaporation in vacuo) including TLC purification of the crude products (EtOAc: CH_2_Cl_2_ = 1: 5; *R*_f_ = 0.83 (**32**), *R*_f_ = 0.58 (**33**)**,**
*R*_f_ = 0.44 (**34**), *R*_f_ = 0.80 (**35**), *R*_f_ = 0.57 (**36**), *R*_f_ = 0.45 (**37**)) gave orange solids of **32** (160 mg, 25%), **33** (223 mg, 40%), **34** (187 mg, 32%), **35** (140 mg, 22%), **36** (190 mg, 35%), and **37** (150 mg, 25%).
Ac–l-Phe–NH–Fn–NH–l-Phe–Boc **(32):** Mp = 145.2 °C. IR (CH_2_Cl_2_) ῠ_max_/cm^−1^: 3430 m (NH_free_), 3302, 3266, 3217 s (NH_assoc_.), 1731 s, 1706 s, 1683 s, 1668 s (C=O_CONH_), 1648 s 1575 s, 1541 s, 1498 s, 1467 s, 1457 s (amide II). IR (KBr) ῠ_max_/cm^−1^: 3294 m (NH_assoc_.), 1658 s, 1651 s (C=O_CONH_), 1568 s, 1533 s, 1499 (amide II). ^1^H-NMR (600 MHz, CDCl_3_) *δ*: 9.21 (1H, s, NH^b^_Fn_), 9.15 (1H, s, NH^a^_Fn_), 7.30–7.17 (11H, m, CH^a and b^_phenyl_ and NH_Ac_), 5.47–5.26 (3H, m, CH-7_Fn_, CH-10_Fn_ and NH_Boc_), 4.84–4.76 (1H, s, CH^a^_α-Phe_), 4.67–4.59 (1H, s, CH^b^_α-Phe_), 4.09–3.88 (6H, m, CH-2_Fn_, CH-3_Fn_, CH-4_Fn_, CH-5_Fn_, CH-8_Fn_ and CH-9_Fn_), 3.19 (1H, dd, *J* = 14.14; 4.46 Hz, CH^a^_β1-Phe_), 3.14 (1H, dd, *J* = 14.89; 5.21 Hz, CH^b^_β1-Phe_), 2.98 (1H, dd, *J* = 13.80; 10.42 Hz, CH^a^_β2-Phe_), 2.89 (1H, dd, *J* = 14.14; 10.05 Hz, CH^b^_β2-Phe_), 2.06 (s, 3H, CH_3-Ac_), 1.43 (9H, s, (CH_3_)_3-Boc_) ppm. ^13^C-NMR (150 MHz, CDCl_3_) *δ*: 171.8 (1C, CO_Ac_), 170.9 (1C, CO^b^_Fn_), 170.7 (1C, CO^a^_Fn_), 157.3 (1C, CO_Boc_), 136.9 (1C, C_q_^a or b^_γ-Phe_), 136.7 (1C, C_q_^a or b^_γ-Phe_), 129.3 (2C, CH^a^^or b^_δ-Phe_), 129.2 (2C, CH^a^^or b^_δ-Phe_), 128.9 (2C, CH^a^^or b^_ε-Phe_), 128.8 (2C, CH^a^^or b^_ε-Phe_), 127.1 (1C, CH^a^^or b^_ζ-Phe_), 127.0 (1C, CH^a^^or b^_ζ-Phe_)_,_ 95.9 (1C, C_q_-6_Fn_), 95.6 (1C, C_q_-1_Fn_), 81.0 (1C, C_qBoc_), 65.9 (2C, CH-8 _Fn_ and CH-9_Fn_), 65.0 (1C, CH-5_Fn_), 64.9 (1C, CH-4_Fn_), 63.1 (1C, CH-7_Fn_), 62.9 (1C, CH-10_Fn_), 61.8 (1C, CH-3_Fn_), 61.5 (1C, CH-2_Fn_), 56.6 (1C, CH^b^_α-Phe_), 56.3 (1C, CH^a^_α-Phe_), 38.1 (1C, CH_2_^b^_β-Phe_), 37.7 (1C, CH_2_^a^_β-Phe_), 28.7 (3C, (CH_3_)_3-Boc_), 23.1 (1C, CH_3-Ac_) ppm. ESI-MS (H_2_O:MeOH= 50:50): *m/z* 653.3 ((M+H)^+^). MALDI-HRMS *m/z* = 652.2337 (calculated for C_35_H_40_N_4_O_5_Fe = 652.2348).Ac–l-Val–NH–Fn–NH–l-Val–Boc **(33):** Mp = 182.4 °C. IR (CH_2_Cl_2_) ῠ_max_/cm^−1^: 3434 m (NH_free_), 3305 s, 3249 m (NH_assoc_.), 1733 s, 1716 s, 1683 s, 166 s (C=O_CONH_), 1637 s, 1571 s, 1540 s, 1505 s, 1458 s (amide II). IR (KBr) ῠ_max_/cm^−1^: 3293 s (NH_assoc_.), 1672 s, 1652 s, 1578 s (C=O_CONH_), 1563 s, 1508 s, 1480 s, 1467 s (amide II). ^1^H-NMR (600 MHz, CDCl_3_) *δ*: 9.20 (1H, s, NH^b^_Fn_), 9.02 (1H, s, NH^a^_Fn_), 6.74 (1H, d, *J* = 7.26 Hz, NH_Ac_), 5.34 (2H, s, CH-7_Fn_ and CH-10_Fn_), 5.20 (1H, d, *J* = 8.38 Hz, NH_Boc_), 4.14 (1H, t, *J* = 7.95 Hz, CH^a^_α-Val_), 4.03 (1H, s, CH-2_Fn_), 3.98–3.94 (3H, m, CH-3_Fn_, CH-4_Fn_ and CH^b^_α-Val_), 3.92 (1H, s, CH-5_Fn_), 3.88 (2H, s, CH-8_Fn_ and CH-9_Fn_), 2.10 (3H, s, CH_3-Ac_), 2.05–1.98 (1H, m, CH^a^_β-Val_), 1.98–1.91 (1H, m, CH^b^_β-Val_), 1.45 (9H, s, (CH_3_)_3-Boc_), 1.04 (3H, d, *J* = 6.60 Hz, CH_3_^a^_γ-Val_), 1.02 (3H, d, *J* = 6.80 Hz, CH_3_^b^_γ-Val_), 0.98 (6H, d, *J* = 6.60 Hz, CH_3_^a^_γ-Val_ and CH_3_^b^_γ-Val_) ppm. ^13^C-NMR (150 MHz, CDCl_3_) *δ*: 171.6 (1C, CO_Ac_), 171.0 (1C, CO^b^_Fn_), 170.7 (1C, CO^a^_Fn_), 157.3 (1C, CO_Boc_), 95.9 (1C, C_q_-6_Fn_), 95.6 (1C, C_q_-1_Fn_), 80.6 (1C, C_qBoc_), 65.7 (2C, CH-8_Fn_ and CH-9_Fn_), 65.1 (1C, CH-5_Fn_), 64.9 (1C, CH-4_Fn_), 62.9 (1C, CH-7_Fn_), 62.8 (1C, CH-10_Fn_), 61.9 (1C, CH-3_Fn_), 61.4 (1C, CH-2_Fn_), 61.3 (1C, CH^b^_α-Val_), 61.1 (1C, CH^a^_α-Val_), 30.5 (1C, CH^b^_β-Val_), 30.3 (1C, CH^a^_β-Val_), 28.7 (3C, (CH_3_)_3-Boc_), 23.4 (1C, CH_3-Ac_), 19.7 (1C, CH_3_^a or b^_γ-Val_), 19.44 (2C, CH_3_^a and b^_γ-Val_), 19.36 (1C, CH_3_^a or b^_γ-Val_) ppm. ESI-MS (H_2_O:MeOH= 50:50): *m/z* 579.3 ((M+Na)^+^). MALDI-HRMS *m/z* = 556.2339 (calculated for C_27_H_40_N_4_O_5_Fe = 556.2348).Ac–l-Leu–NH–Fn–NH–l-Leu–Boc **(34):** Mp = 206.6 °C. IR (CH_2_Cl_2_) ῠ_max_/cm^−1^: 3434 m (NH_free_), 3301 s, 3253 s (NH_assoc_.), 1733 s, 1684 s, 1668 s (C=O_CONH_), 1653 s, 1573 s, 1540 s, 1506 s, 1457 s (amide II). IR (KBr) ῠ_max_/cm^−1^: 3281 s (NH_assoc_.), 1667 s, 1650 s (C=O_CONH_), 1568 s, 1530 s, 1484 s (amide II). ^1^H-NMR (600 MHz, CDCl_3_) *δ*: 9.36 (1H, s, NH^b^_Fn_), 9.13 (1H, s, NH^a^_Fn_), 7.28 (1H, d, *J* = 6.20 Hz, NH_Ac_), 5.35 (1H, s, CH-7_Fn_), 5.34 (1H, s, CH-10_Fn_), 5.14 (1H, d, *J* = 8.02 Hz, NH_Boc_), 4.49–4.43 (1H, m, CH^a^_α-Leu_), 4.31–4.25 (1H, m, CH^b^_α-Leu_), 4.05 (1H, s, CH-3_Fn_), 3.98 (1H, s, CH-4_Fn_), 3.95 (1H, s, CH-2_Fn_), 3.90 (1H, s, CH-5_Fn_), 3.87 (2H, s, CH-8_Fn_ and CH-9_Fn_), 2.09 (3H, s, CH_3-Ac_), 1.79–1.65 (3H, m, CH^a^_β1-Leu_, CH^a^_γ-Leu_ and CH^b^_γ-Leu_), 1.59–1.50 (3H, m, CH^a^_β2-Leu_, CH^b^_β1-Leu_ and CH^b^_β2-Leu_), 1.45 (9H, s, (CH_3_)_3-Boc_), 0.92 (6H, t, *J* = 6.80 Hz, CH_3_^b^_δ1-Leu_ and CH_3_^b^_δ2-Leu_), 0.86 (6H, d, *J* = 6.58 Hz, CH_3_^a^_δ1-Leu_ and CH_3_^a^_δ2-Leu_) ppm. ^13^C-NMR (150 MHz, CDCl_3_) *δ*: 172.0 (1C, CO_Ac_), 171.75 (1C, CO^b^_Fn_), 171.67 (1C, CO^a^_Fn_), 157.3 (1C, CO_Boc_), 96.2 (1C, C_q_-6_Fn_), 95.9 (1C, C_q_-1_Fn_), 80.7 (1C, C_qBoc_), 65.7 (2C, CH-8_Fn_ and CH-9_Fn_), 64.9 (1C, CH-5_Fn_), 64.7 (1C, CH-4_Fn_), 62.9 (1C, CH-7_Fn_), 62.7 (1C, CH-10_Fn_), 61.6 (1C, CH-3_Fn_), 61.3 (1C, CH-2_Fn_), 54.1 (1C, CH^b^_α-Leu_), 53.9 (1C, CH^a^_α-Leu_), 41.1 (1C, CH_2_^b^_β-Leu_), 40.6 (1C, CH_2_^a^_β-Leu_), 28.7 (3C, (CH_3_)_3-Boc_), 25.0 (1C, CH^b^_γ-Leu_), 24.9 (1C, CH^a^_γ-Leu_), 23.5 (1C, CH_3-Ac_), 23.1, 23.0, 21.8 and 21.7 (4C, CH_3_^a^_δ1- and δ2-Leu_ and CH_3_^b^ _δ1- and δ2-Leu_) ppm. ESI-MS (H_2_O:MeOH= 50:50): *m/z* 607.3 ((M+Na)^+^). MALDI-HRMS *m/z* = 584.2641 (calculated for C_29_H_44_N_4_O_5_Fe = 584.2661).Ac–d-Phe–NH–Fn–NH–d-Phe–Boc (**35):** Mp = 140.1 °C. IR (CH_2_Cl_2_) ῠ_max_/cm^−1^: 3432 m (NH_free_), 3301 s (NH_assoc_.), 1731 s, 1682 s, 1668 s (C=O_CONH_), 1571 s, 1498 s, 1455 s, 1442 s (amide II). IR (KBr) ῠ_max_/cm^−1^: 3293 s (NH_assoc_.), 1657 s, 1651 s (C=O_CONH_), 1570 s, 1547 m, 1531 m, 1498 m, 1491 m, 1454 s (amide II). ^1^H-NMR (600 MHz, CDCl_3_) *δ*: 9.18 (1H, s, NH^b^_Fn_), 9.12 (1H, s, NH^a^_Fn_), 7.30–7.17 (10H, m, CH^a and b^_phenyl_), 7.07 (1H, s, NH_Ac_), 5.36 (1H, s, CH-7_Fn_), 5.34 (1H, s, CH-10_Fn_), 5.30 (1H, s, *J* = 5.88 Hz, NH_Boc_), 4.79 (1H, s, CH^a^_α-Phe_), 4.62 (1H, s, CH^b^_α-Phe_), 4.08–3.89 (6H, m, CH-2_Fn_, CH-3_Fn_, CH-4_Fn_, CH-5_Fn_, CH-8_Fn_ and CH-9_Fn_), 3.21–3.10 (2H, m, CH^a^_β1-Phe_ and CH^b^_β1-Phe_), 2.97 (1H, t, *J* = 10.91 Hz, CH^a^_β2-Phe_), 2.88 (1H, t, *J* = 10.91 Hz, CH^a^_β2-Phe_), 2.05 (3H, s, CH_3-Ac_), 1.42 (9H, s, (CH_3_)_3-Boc_) ppm. ^13^C-NMR (150 MHz, CDCl_3_) *δ*: 171.8 (1C, CO_Ac_), 170.9 (1C, CO^b^_Fn_), 170.6 (1C, CO^a^_Fn_), 157.3 (1C, CO_Boc_), 136.9 (1C, C_q_^a or b^_γ-Phe_), 136.7 (1C, C_q_^a or b^_γ-Phe_), 129.3 (2C, CH^a or b^_δ-Phe_), 129.2 (2C, CH^a or b^_δ-Phe_), 128.9 (2C, CH^a or b^_ε-Phe_), 128.8 (2C, CH^a or b^_ε-Phe_), 127.2 (1C, CH^a or b^_ζ-Phe_), 127.1 (1C, CH^a or b^_ζ-Phe),_ 95.9 (1C, C_q_-6_Fn_), 95.5 (1C, C_q_-1_Fn_), 81.0 (1C, C_qBoc_), 65.9 (2C, CH-8 _Fn_ and CH-9_Fn_), 65.0 (1C, CH-5_Fn_), 64.8 (1C, CH-4_Fn_), 63.1 (1C, CH-7_Fn_), 62.9 (1C, CH-10_Fn_), 61.9 (1C, CH-3_Fn_), 61.5 (1C, CH-2_Fn_), 56.6 (1C, CH^b^_α-Phe_), 56.3 (1C, CH^a^_α-Phe_), 38.1 (1C, CH_2_^b^_β-Phe_), 37.7 (1C, CH_2_^a^_β-Phe_), 28.7 (3C, (CH_3_)_3-Boc_), 23.1 (1C, CH_3-Ac_) ppm. ESI-MS (H_2_O:MeOH= 50:50): *m/z* 653.1 ((M+H)^+^), calculated for C_35_H_40_N_4_O_5_Fe = 652.2348.Ac–d-Val–NH–Fn–NH–d-Val–Boc **(36):** Mp = 201.4 °C. IR (CH_2_Cl_2_) ῠ_max_/cm^−1^: 3435 m (NH_free_), 3306 s (NH_assoc_.), 1731 s, 1681 s, 1667 s (C=O_CONH_), 1571 s, 1503 s (amide II). IR (KBr) ῠ_max_/cm^−1^: 3287 s (NH_assoc_.), 1671 s, 1652 s (C=O_CONH_), 1577 s, 1564 s, 1508 s (amide II). ^1^H-NMR (600 MHz, CDCl_3_) *δ*: 9.21 (1H, s, NH^b^_Fn_), 9.05 (1H, s, NH^a^_Fn_), 6.78 (1H, d, *J* = 6.91 Hz, NH_Ac_), 5.36 (2H, s, CH-7_Fn_ and CH-10_Fn_), 5.21 (1H, d, *J* = 8.41 Hz, NH_Boc_), 4.16 (1H, t, *J* = 8.04 Hz, CH^a^_α-Val_), 4.04 (1H, s, CH-2_Fn_), 4.01–3.96 (3H, m, CH-3_Fn_, CH-4_Fn_ and CH^b^_α-Val_), 3.94 (1H, s, CH-5_Fn_), 3.90 (2H, s, CH-8_Fn_ and CH-9_Fn_), 2.11 (3H, s, CH_3-Ac_), 2.04 (1H, q, *J* = 6.96 Hz, CH^a^_β-Val_), 1.97 (1H, q, *J* = 6.96 Hz, CH^b^_β-Val_), 1.47 (9H, s, (CH_3_)_3-Boc_), 1.07 (3H, d, *J* = 6.69 Hz, CH_3_^a^_γ-Val_), 1.04 (3H, d, *J* = 6.69 Hz, CH_3_^b^_γ-Val_), 1.00 (6H, d, *J* = 6.69 Hz, CH_3_^a^_γ-Val_ and CH_3_^b^_γ-Val_) ppm. ^13^C-NMR (150 MHz, CDCl_3_) *δ*: 171.5 (1C, CO_Ac_), 171.0 (1C, CO^b^_Fn_), 170.7 (1C, CO^a^_Fn_), 157.3 (1C, CO_Boc_), 95.9 (1C, C_q_-6_Fn_), 95.6 (1C, C_q_-1_Fn_), 80.6 (1C, C_qBoc_), 65.7 (2C, CH-8_Fn_ and CH-9_Fn_), 65.1 (1C, CH-5_Fn_), 64.9 (1C, CH-4_Fn_), 62.9 (1C, CH-7_Fn_), 62.8 (1C, CH-10_Fn_), 61.8 (1C, CH-3_Fn_), 61.4 (1C, CH-2_Fn_), 61.2 (1C, CH^b^_α-Val_), 61.1 (1C, CH^a^_α-Val_), 30.4 (1C, CH^b^_β-Val_), 30.2 (1C, CH^a^_β-Val_), 28.7 (3C, (CH_3_)_3-Boc_), 23.4 (1C, CH_3-Ac_), 19.7 (1C, CH_3_^a or b^_γ-Val_), 19.44 (2C, CH_3_^a and b^_γ-Val_), 19.38 (1C, CH_3_^a or b^_γ-Val_) ppm. ESI-MS (H_2_O:MeOH= 50:50): *m/z* 579.1 ((M+Na)^+^), calculated for C_27_H_40_N_4_O_5_Fe = 556.2348.Ac–d-Leu–NH–Fn–NH–d-Leu–Boc **(37):** Mp = 207.8 °C. IR (CH_2_Cl_2_) ῠ_max_/cm^−1^: 3434 m (NH_free_), 3292 s (NH_assoc_.), 1733 w, 1663 s (C=O_CONH_), 1492 s (amide II). IR (KBr) ῠ_max_/cm^−1^: 3287 s (NH_assoc_.), 1683 s, 1667 s, 1651 s (C=O_CONH_), 1565 m, 1526 m, 1483 m (amide II). ^1^H-NMR (600 MHz, CDCl_3_) *δ*: 9.36 (1H, s, NH^b^_Fn_), 9.14 (1H, s, NH^a^_Fn_), 7.25 (1H, s, NH_Ac_), 5.38 (2H, s, CH-7_Fn_ and CH-10_Fn_), 5.15 (1H, d, *J* = 7.77 Hz, NH_Boc_), 4.51–4.41 (1H, m, CH^a^_α-Leu_), 4.32–4.24 (1H, m, CH^b^_α-Leu_), 4.13–3.83 (6H, m, CH-2_Fn_, CH-3_Fn_, CH-4_Fn_, CH-5_Fn_, CH-8_Fn_ and CH-9_Fn_), 2.09 (3H, s, CH_3-Ac_), 1.79–1.64 (3H, m, CH^a^_β1-Leu_, CH^a^_γ-Leu_ and CH^b^_γ-Leu_), 1.60–1.49 (3H, m, CH^a^_β2-Leu_, CH^b^_β1-Leu_ and CH^b^_β2-Leu_), 1.45 (9H, s, (CH_3_)_3-Boc_), 0.92 (6H, t, *J* = 6.62 Hz, CH_3_^b^_δ1-Leu_ and CH_3_^b^_δ2-Leu_), 0.86 (6H, d, *J* = 6.49 Hz, CH_3_^a^_δ1-Leu_ CH_3_^a^_δ2-Leu_) ppm. ^13^C-NMR (150 MHz, CDCl_3_) *δ*: 171.9 (1C, CO_Ac_), 171.6 (2C, CO^a^_Fn_ and CO^b^_Fn_), 157.2 (1C, CO_Boc_), 96.2 (1C, C_q_-6_Fn_), 95.9 (1C, C_q_-1_Fn_), 80.7 (1C, C_qBoc_), 65.7 (2C, CH-8_Fn_ and CH-9_Fn_), 65.0 (1C, CH-5_Fn_), 64.8 (1C, CH-4_Fn_), 62.9 (1C, CH-7_Fn_), 62.7 (1C, CH-10_Fn_), 61.5 (1C, CH-2_Fn_), 61.3 (1C, CH-3_Fn_), 54.0 (1C, CH^b^_α-Leu_), 53.9 (1C, CH^a^_α-Leu_), 41.0 (1C, CH_2_^b^_β-Leu_), 40.5 (1C, CH_2_^a^_β-Leu_), 28.7 (3C, (CH_3_)_3-Boc_), 25.0 (1C, CH^b^_γ-Leu_), 24.8 (1C, CH^a^_γ-Leu_), 23.4 (1C, CH_3-Ac_), 23.1, 23.0, 21.7 and 21.6 (4C, CH_3_^a^_δ1- and δ2-Leu_ and CH_3_^b^ _δ1- and δ2-Leu_) ppm. ESI-MS (H_2_O:MeOH = 50:50): *m/z* 607.2 ((M+Na)^+^), calculated for C_29_H_44_N_4_O_5_Fe = 584.2661.

#### 3.1.2. Computational Details

A conformational search was performed on the l-series of each enantiomeric pair (compounds **32**–**34**). The same Boltzmann distribution of conformers is expected for the d-series. At first, each compound was subjected to a series of low-level optimizations with molecular mechanics, OPLS2005 force field, in MacroModel v10.3 [52,53]. A set of the most stable conformers were further optimized at a high level of theory (B3LYP/Lanl2DZ) in Gaussian16 [91] with default convergence criteria. At last, only a few of the most stable conformers were modeled at the B3LYP-D3/6-311+G(d,p) (LanL2DZ basis set on Fe) level of theory [92,93,94] while surrounding solvent (chloroform) was described as a polarizable continuum (SMD) [95]. Vibrational analysis was performed to verify each structure as a minimum on the potential energy surface. The reported energies refer to standard Gibbs free energies at 298 K. AIMAll package and QTAIM theory were used to characterize hydrogen bonds [96]. Topological parameters of the displayed bond critical points between hydrogen bond acceptors and hydrogen atoms were calculated and verified according to the Koch and Popelier criteria [54].

#### 3.1.3. Crystallographic Study

X-ray diffraction: data collection for compound **33** was carried out on an Enraf-Nonius CAD4 diffractometer equipped with an Oxford Cryosystems Cryostream Series 700 liquid nitrogen cooling device. The WinGX standard procedure was applied for data reduction [97,98]. Three standard reflections were measured every 120 min as an intensity control. Since the absorption coefficients were low and the crystals small (Table 4), no absorption correction was applied.

Data collection for compound **36** was performed on a Rigaku Oxford Diffraction Synergy S diffractometer (dual Cu/Mo microsource) equipped with an Oxford Instruments CryoJet liquid nitrogen cooling device. Program package CrysAlis PRO [99] was used for data reduction and numerical absorption correction.

The structures were solved using SHELXS97 [100] and refined with SHELXL-2017 [101]. Models were refined using the full-matrix least squares refinement; all non-hydrogen atoms were refined anisotropically. Hydrogen atoms were located in a difference Fourier map and refined either as riding entities or a mixture of free restrained and riding entities.

Molecular geometry calculations were performed by PLATON [102] and molecular graphics were prepared using ORTEP-3 [103], and Mercury [104]. Crystallographic and refinement data for the structures reported in this paper are shown in Table 4.

#### 3.1.4. Biological Activity


Antitumor activity


Materials: trypsin-EDTA (0.25%), FBS (fetal bovine serum), DMEM (Dulbecco’s Modified Eagle Medium), RNAse A, insulin, glutamine, penicillin, and streptomycin were purchased from Sigma-Aldrich. PBS (phosphate buffer saline) was prepared from the following ingredients: sodium chloride was purchased from Honeywell Research Chemicals, calcium chloride was purchased from Gram-Mol, disodium phosphate and monopotassium phosphate were purchased from Merck. Accutase Cell Detachment Solution was purchased from Capricorn Scientific GmbH (Ebsdorfergrund, Germany). Annexin V-FITC and propidium iodide were purchased from BD Biosciences, while Annexin Binding Buffer was prepared from the following ingredients: HEPES was purchased from Carl Roth, and calcium chloride and sodium hydroxide were purchased from Gram-Mol. Absolute ethanol was purchased from Merck.

Cell culture: The HeLa (cervical adenocarcinoma), MCF-7 (breast adenocarcinoma), and HepG2 (hepatocellular carcinoma) cells were cultured as monolayers and maintained in DMEM supplemented with 10% (*v*/*v*) FBS, 2 mmol/L l-glutamine, 100 U/mL penicillin, and 100 mg/mL streptomycin in a humidified atmosphere with 5% CO_2_ at 37 °C. Media for MCF-7 were additionally supplemented with 0.01 mg/mL human recombinant insulin. Cells were washed in PBS and detached with 0.25% (*v*/*v*) Trypsin-EDTA solution.

The growth inhibition activity was assessed according to the slightly modified procedure performed at the National Cancer Institute, Developmental Therapeutics Program [85]. The cells were inoculated onto standard 96-well microtiter plates on day 0. Cell concentrations were adjusted according to their respective growth rates: 3000 cells per well for HeLa and HepG2, and 5000 cells per well for MCF-7. Test agents were then added the next day in five dilutions (5, 10, 50, 100, and 350 umol/L) and incubated over a further 72 h. Working dilutions were freshly prepared on the day of testing. The solvent (EtOH) was also tested for possible inhibitory activity at the same concentration as in tested solutions. After 72 h of incubation, the cell growth rate was evaluated by the MTT assay, which detects dehydrogenase activity in viable cells [23,105] The absorbance (OD, optical density) was measured on a microplate reader at 595 nm. Percentage of growth (PG) of the cell lines was calculated using one of the following two expressions:
If (mean OD_test_ − mean OD_tzero_) ≥ 0, then: PG = 100 × (mean OD_test_ − mean OD_tzero_)/(mean OD_ctrl_ − mean OD_tzero_)
If (mean OD_test_ − mean OD_tzero_) < 0, then: PG = 100 × (mean OD_test_ − mean OD_tzero_)/OD_tzero_
where mean OD_tzero_ = the average of optical density measurements before exposure of cells to the test compound; mean OD_test_ = the average of optical density measurements after the desired period of time; mean OD_ctrl_ = the average of optical density measurements after the desired period of time without exposure of cells to the test compound [104]. Each test point was performed in quadruplicate in three individual experiments. The results are expressed as IC_50_, which is the concentration necessary for 50% inhibition. The IC_50_ values for each compound are calculated from dose response curves using linear regression analysis by fitting the test concentrations that give PG values above and below the reference value (i.e., 50%). If, however, for a given cell line all of the tested concentrations produce PGs exceeding the respective reference level of effect (e.g., PG value of 50), then the highest tested concentration is assigned as the default value, which is preceded by a sign >. Each result is the mean value from three separate experiments.

Based on the IC_50_ values determined by the MTT assay, compound **37** was selected for further analysis of apoptosis and cell cycle.

Treatment: for apoptosis, 10^5^ cells per well were seeded in 12-well plates, while for the cell cycle, 2 × 10^5^ cells per well were seeded in 6-well plates. Once the cell monolayers reached 80% of confluency, the cells were treated for 24 h with four different concentrations of the test compound (26 µM, 44 µM, 61 µM, and 105 µM) dissolved in DMEM completed medium. For both apoptosis and cell cycle, the medium containing ethanol without the tested compound was used as a negative control.

Apoptosis analysis by flow cytometry: after the 24-h treatment, the floating HeLa cells were collected while the attached cells were washed with PBS and detached with Accutase. Both attached and detached cells were pooled together, centrifuged, resuspended in 1× Annexin Binding Buffer, and stained with Annexin V-FITC and PI according to the manufacturer’s instructions. Samples were analyzed using the Navios^TM^ flow cytometer (Beckman Coulter Life Sciences, Miami, FL, USA) to detect viable (Annexin V-FITC-negative/PI-negative), total apoptotic (Annexin V-FITC-positive/ PI-negative and Annexin V-FITC-positive/PI-positive), and necrotic cells (Annexin V-FITC-negative/PI-positive). Ten thousand cells were analyzed per sample. Data were analyzed using the FlowLogic software (Inivai, Melbourne, Australia). Analysis was performed on two biological replicates, with each concentration of compound **37** tested in duplicate (Appendix A).

Cell cycle analysis by flow cytometry: after the 24-h treatment, the floating HeLa cells were collected, while the attached cells were washed with PBS and detached with Accutase. Both attached and detached cells were pooled together, centrifuged, and washed twice with PBS. After washing, cells were resuspended in 1.5 mL PBS and fixed by adding two volumes of the ice-cold absolute ethanol. After a minimum of 72 h of fixation, cells were centrifuged, washed twice in PBS, and then incubated with RNase A (0.1 mg/mL) and PI (50 µg/mL) at room temperature for 30 min in the dark. The cell cycle was analyzed using the DxFLEX flow cytometer (Beckman Coulter Life Sciences, Miami, FL, USA). Twenty thousand cells were analyzed per sample. FlowLogic software (Inivai, Melbourne, Australia) was used to determine the percentage of cells in each phase of the cell cycle (G0/G1, S, G2/M). Each concentration of compound **37** was tested in duplicate and each experiment was performed twice.

Statistical analysis: the obtained results were analyzed using GraphPad Prism 9 software. Data are presented as mean value ± SEM of two independent experiments performed in duplicate. For comparison between the control and treated groups, one-way ANOVA test, followed by post-hoc Tukey’s test, was used. Statistical significance was considered at a *p* value < 0.05.
Antimicrobial activity

Microorganisms: To evaluate the antimicrobial properties of the tested peptides, thirteen microorganisms were used: Gramme-positive bacteria (*Staphylococcus aureus*, *Bacillus subtilis*, *Enterococcus faecium*, *Listeria monocytogenes*), Gramme-negative bacteria (*Pseudomonas aeruginosa*, *Escherichia coli*, *Salmonella enterica* s. Typhimurium), lactic acid bacteria (*Leuconostoc mesenteroides*, *Lactobacillus plantarum*) and yeasts (*Candida albicans*, *Candida utilis*, *Rhodotorula* sp. and *Saccharomyces cerevisiae*). The microorganisms were stored on slant agar in the microorganism collection of the Laboratory of General Microbiology and Food Microbiology and the Laboratory of Fermentation and Yeast Technology of the Faculty of Food Technology and Biotechnology, University of Zagreb (Croatia). Antimicrobial activity tests: In the first step, the antimicrobial activities of the tested peptides were determined using the disc diffusion method to verify the efficacy of the samples on all tested strains. After overnight growth of cultures under anaerobic conditions at 37 °C in Muler Hinton broth (Gramme-positive and Gramme-negative bacteria), MRS broth at 32 °C (lactic acid bacteria), and Muler Hinton broth with 2% glucose at 28 °C (yeast), cell density was adjusted to 0.5 McFarland using a spectrophotometer (A_550 nm_ ≈ 0.125). Agar plates were then inoculated directly from the suspension, and the inoculum was spread with a sterile swab according to the CLSI protocol. Ferrocene peptides were diluted in DMSO (0.1 g/mL). Philtre paper discs were then placed on the inoculated medium using flamed forceps and the ferrocene samples (10 µL) were applied to the discs (diameter = 6 mm; 1mg/disc). Kanamycin and nystatin were used as positive controls and DMSO as a negative control. Petri dishes were incubated anaerobically, and zones of inhibition were measured with a ruler after 24 h Appendix A in the Appendix A). For each peptide and microorganism, the experiments were performed in duplicate.
Antioxidant activity assays

The methods used to determine the antioxidant activity are based on the study of a reaction in which a free radical is generated and inhibited by the addition of the sample whose antioxidant activity is being measured. DPPH radical scavenging activity and ferric ion reducing antioxidant power (FRAP) are used to determine the antioxidant activity of the tested peptides (*c* = 1 mM).

DPPH assays: the DPPH radical scavenging assay (1,1-diphenyl-2-picrylhydrazyl) was performed according to the method of Brand-Williams et al. [106]. Ethanol (1 mL) and 0.1 mM DPPH working solution (2 mL) were added to the samples (150 µL). The absorbance at 525 nm was measured after 30 min of incubation in the dark. The DPPH reagent and ethanol were used as blank reference, and 0.1 mM and 0.5 mM Trolox were used to compare the percentage of DPPH radical inhibition. The percentage of DPPH inhibition was calculated using the following formula: % of DPPH reduction = [Ao − As]/Ao × 100, where Ao is the absorbance of DPPH solution with ethanol and As is the absorbance of a DPPH solution with the sample. Results are expressed as percent inhibition of 0.1 mM Trolox. All experiments were performed in triplicate.

FRAP assay: the FRAP assay was performed as previously described [107], with some modifications. FRAP reagent solution was prepared from the mixture of acetate buffer (300 mM, pH = 3.6), TPTZ (10 mM solution of TPTZ in 40 mM HCl) and FeCl_3_ × 6H_2_O (20 mM) in a volume ratio of 10:1:1). The working reagent FRAP was freshly prepared. All samples, standards , and reagents were pre-incubated at 37 ^°^C. The sample to be analyzed (80 µL) was mixed with FRAP reagent (2080 µL) and distilled water (240 µL) at 37 °C. After 5 min, the absorbance was measured at 593 nm. The standard curve was generated using a serial dilution (0.1–2.0 mM) of the stock Trolox solution. The results were expressed as mM Trolox equivalents. All experiments were also performed in triplicate.

## 4. Conclusions

A common protocol involving synthesis, in-depth experimental (IR, NMR, CD), computational (DFT) conformational analysis, and biological evaluation (MTT, cell cycle, apoptosis, disk diffusion, DPPH, FRAP) can provide valuable information on structural properties and antitumor, antimicrobial, and antioxidant potential.

Homochiral conjugates of ferrocene-1,1′-diamine with l-/d-Phe (**32/35**), l-/d-Val (**33/36**), and l-/d-Leu (**34/37**) form a very robust structural motif consisting of two intramolecularly hydrogen-bonded 10-membered rings (two β-turns). A ferrocene core serves as a template for the correct adjustment of two *N*-protected amino acids attached to the opposite cyclopentadienyl rings in condensed phases, in solution, and in the solid state. By achieving the same intramolecular hydrogen-bonding pattern and minimizing the effects resulting from different conformations, the biological activity of the investigated compounds can be uniquely ascribed either to the different side chains of the chiral amino acids (Phe, Val, Leu) or to the opposite central chirality of each amino acid. The antitumor activity of the homochiral peptides tested suggests that the amino acid side chain is more important for the inhibitory effect compared to the absolute configuration. The preliminary results of biological evaluation of d-Leu conjugate **37** indicate that this compound is associated with the induction of G0/G1 phase cell cycle arrest in the HeLa cell line. Further studies are required to investigate the exact mechanism underlying this effect.

In order to obtain a small library of ferrocene peptidomimetics with a defined structure–activity relationship, our further studies will be directed to the conformational and biological evaluation of the heterochiral analogues of the peptides studied here, whose preliminary tests indicate a pronounced biological potential.

## Data Availability

The crystallographic data have been deposited in the Cambridge Structural Database as entries no. 2175174 and 2175175. The DFT, spectroscopic, and biological evaluation data are provided as figures and tables and are included in this paper.

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
