# Peer review of "Hydrogen Bonding Drives Helical Chirality via 10-Membered Rings in Dipeptide Conjugates of Ferrocene-1,1′-Diamine"

_ijms, 2022, doi:10.3390/ijms232012233_

Round 1

Reviewer 1 Report

This work presents the synthesis and characterization of homochiral conjugates of ferrocene-1,1'-diamine  with L-/D-Phe, L-/D-Val and L-/D-Leu. Their characterization includes spectroscopic data ( IR, NMR, CD), an X-ray crystal structure analysis and a conformational study using density functional theory, as well as an antitumor, antimicrobial and antioxidant potential activity analysis.

There is a lot of work that the authors have been done that it is well presented in the manuscript. I have only a couple of minor comments that I hope would improve the quality of the paper.

Although the theoretical conformational analysis shows that the most stable conformers correspond to the ones with double intramolecular hydrogen bonds / beta-turns, it does not discard that conformations with no hydrogen bonds / beta turns are closer in energy. The energy gap between those to set of structures need to be computed and presented properly in the manuscript. The goal here is to see how easily these compounds loose their beta-turn shape. All data presented in the SI regarding the calculations are minimal and they need to be improved.

The authors need to be more careful include proper citations for the theoretical methods that are using. For instance, DFT functionals and basis set need be properly cited.

Overall this work deserves to be pusblished in IJMS.

Author Response

Dear Editor,

Thank you for giving us the opportunity to submit a revised draft of our manuscript entitled "Hydrogen bonding drives helical chirality via 10-membered rings in dipeptide conjugates of ferrocene-1,1’-diamine" for the special issue "Recent Advance in Hydrogen Bonding" in the International Journal of Molecular Sciences.

We are grateful to you and the reviewers for the time and effort you took to evaluate our manuscript. The revisions we have made are highlighted in yellow in the manuscript and supplementary material.

We address the reviewers’ comments and concerns point by point below.

Sincerely,

Lidija Barišić

Reviewer 1

Comment 1: This work presents the synthesis and characterization of homochiral conjugates of ferrocene-1,1'-diamine with L-/D-Phe, L-/D-Val and L-/D-Leu. Their characterization includes spectroscopic data (IR, NMR, CD), an X-ray crystal structure analysis and a conformational study using density functional theory, as well as an antitumor, antimicrobial and antioxidant potential activity analysis.

There is a lot of work that the authors have been done that it is well presented in the manuscript.

I have only a couple of minor comments that I hope would improve the quality of the paper. Although the theoretical conformational analysis shows that the most stable conformers correspond to the ones with double intramolecular hydrogen bonds / beta-turns, it does not discard that conformations with no hydrogen bonds / beta turns are closer in energy. The energy gap between those to set of structures need to be computed and presented properly in the manuscript. The goal here is to see how easily these compounds loose their beta-turn shape.

Response: This is indeed a very good observation. Double ten-membered rings are very robust HB motifs and they are an integral part of the most stable conformers, regardless of the amino acids included in the substituents. The other families of conformers with somewhat different distribution of hydrogen bonds are about 4-5 kJ mol-1 or more above the energy calculated for the most stable conformer in each series, but we have not included them in this analysis because of simplicity. As suggested by Reviewer, the energies of the “open structures” with no hydrogen bonds are now calculated and the corresponding relative energies are further discussed in the text (p. 6, lines 238-243). Figures of these geometries are now included in the Supplementary Material (p. 9, Figure S1 b).

Comment 2: All data presented in the SI regarding the calculations are minimal and they need to be improved.

Response: Among the already mentioned data (energy levels of the most stable conformers, and geometrical properties of the hydrogen bonds) in the revised version of the manuscript, the geometries of the “open conformers” are now represented as well (Figure S1 b). The Cartesian coordinates of the most stable conformers have also been added to the Supporting information (p. 10, Table S1).

Comment 3: The authors need to be more careful include proper citations for the theoretical methods that are using. For instance, DFT functionals and basis set need be properly cited.

Response: Proper citations (ref. 95-97) of the functional and basis sets are now added to the text.

Comment 2: Overall this work deserves to be pusblished in IJMS.

Response: We would like to thank you for your constructive comments and suggestions which helped us to improve the quality of our paper.

Dear Editor,

Thank you for giving us the opportunity to submit a revised draft of our manuscript entitled "Hydrogen bonding drives helical chirality via 10-membered rings in dipeptide conjugates of ferrocene-1,1’-diamine" for the special issue "Recent Advance in Hydrogen Bonding" in the International Journal of Molecular Sciences.

We are grateful to you and the reviewers for the time and effort you took to evaluate our manuscript. The revisions we have made are highlighted in yellow in the manuscript and supplementary material.

We address the reviewers’ comments and concerns point by point below.

Sincerely,

Lidija Barišić

Reviewer 1

Comment 1: This work presents the synthesis and characterization of homochiral conjugates of ferrocene-1,1'-diamine with L-/D-Phe, L-/D-Val and L-/D-Leu. Their characterization includes spectroscopic data (IR, NMR, CD), an X-ray crystal structure analysis and a conformational study using density functional theory, as well as an antitumor, antimicrobial and antioxidant potential activity analysis.

There is a lot of work that the authors have been done that it is well presented in the manuscript.

I have only a couple of minor comments that I hope would improve the quality of the paper. Although the theoretical conformational analysis shows that the most stable conformers correspond to the ones with double intramolecular hydrogen bonds / beta-turns, it does not discard that conformations with no hydrogen bonds / beta turns are closer in energy. The energy gap between those to set of structures need to be computed and presented properly in the manuscript. The goal here is to see how easily these compounds loose their beta-turn shape.

Response: This is indeed a very good observation. Double ten-membered rings are very robust HB motifs and they are an integral part of the most stable conformers, regardless of the amino acids included in the substituents. The other families of conformers with somewhat different distribution of hydrogen bonds are about 4-5 kJ mol-1 or more above the energy calculated for the most stable conformer in each series, but we have not included them in this analysis because of simplicity. As suggested by Reviewer, the energies of the “open structures” with no hydrogen bonds are now calculated and the corresponding relative energies are further discussed in the text (p. 6, lines 238-243). Figures of these geometries are now included in the Supplementary Material (p. 9, Figure S1 b).

Comment 2: All data presented in the SI regarding the calculations are minimal and they need to be improved.

Response: Among the already mentioned data (energy levels of the most stable conformers, and geometrical properties of the hydrogen bonds) in the revised version of the manuscript, the geometries of the “open conformers” are now represented as well (Figure S1 b). The Cartesian coordinates of the most stable conformers have also been added to the Supporting information (p. 10, Table S1).

Comment 3: The authors need to be more careful include proper citations for the theoretical methods that are using. For instance, DFT functionals and basis set need be properly cited.

Response: Proper citations (ref. 95-97) of the functional and basis sets are now added to the text.

Comment 2: Overall this work deserves to be pusblished in IJMS.

Response: We would like to thank you for your constructive comments and suggestions which helped us to improve the quality of our paper.

Reviewer 2 Report

The article presented by Kovačević and co-workers presents a ferrocene obtention and characterization with a clear orientation towards its applicability in therapeutics.

As positive points of this report, the correct and exhaustive characterization of the compounds obtained should be highlighted. The chemical part of the manuscript includes results and discussion well supported by experimental data. Only, this reviewer recommends to the authors in this section an improvement in Figure 3, where the panels are not well aligned and marked, and should appear (a), (b) and (c) in the upper left part of the panel so that it does not lead to error in the denomination.

However, the biological assays part of the report does not seem to contribute much to the manuscript.

The authors present 3 activity assays: antitumor, antimicrobial and antioxidant.

In the antitumor part, the authors present an evaluation of cytotoxicity. In this case, the graphs presented by the authors are striking, since the incubation time at which the viability measurements were taken does not appear, and the viability values above 100 % are very surprising. Also, the appearance of negative viability values in the assays in the MCF-7 line. In the case that these values were correct, the authors should explain the reason for these values or if it is due to the methodology used. In this case, the methodology and the associated calculation should be revised. 

It appears that compounds 32 and 35 are the most active in the MCF7 line, 37 the most active in the HeLa line with the Hep62 line being insensitive to these compounds. Despite obtaining more interesting values in the MCF-7 line, the authors have continued working with the Hela cell line for the following assays.

The apoptosis and cell cycle assays provided do not obtain conclusive results of any kind. Neither the tested compound generates a relevant apoptosis nor a sufficient cell cycle arrest to qualify as "antitumor" to correlate with the presented cytotoxicity. Clearly, this is a preliminary assessment, but the authors should consider what these results add to their manuscript. In the case of assays performed by flow cytometry, I would recommend that the authors include the cytometry graphs with an analysis performed to justify the graphs presented. Similarly, subsequent activity (antioxidant activity and antimicrobial) assays also contribute nothing relevant to the compounds. 

Authors must consider presenting, for example, the same assays for compounds 32 and 35 in the MCF-7 line. If better results are collected for these compounds, it will be of interest to include the studies in the manuscript.

I recommend a major revision of the article before acceptance of the report for publication.

Author Response

Dear Editor,

Thank you for giving us the opportunity to submit a revised draft of our manuscript entitled "Hydrogen bonding drives helical chirality via 10-membered rings in dipeptide conjugates of ferrocene-1,1’-diamine" for the special issue "Recent Advance in Hydrogen Bonding" in the International Journal of Molecular Sciences.

We are grateful to you and the reviewers for the time and effort you took to evaluate our manuscript. The revisions we have made are highlighted in yellow in the manuscript and supplementary material.

We address the reviewers’ comments and concerns point by point below.

Sincerely,

Lidija Barišić

Reviewer 2

Comment 1: The article presented by Kovačević and co-workers presents a ferrocene obtention and characterization with a clear orientation towards its applicability in therapeutics.

As positive points of this report, the correct and exhaustive characterization of the compounds obtained should be highlighted. The chemical part of the manuscript includes results and discussion well supported by experimental data. Only, this reviewer recommends to the authors in this section an improvement in Figure 3, where the panels are not well aligned and marked, and should appear (a), (b) and (c) in the upper left part of the panel so that it does not lead to error in the denomination.

Response: We have revised Figure 3 as recommended by Reviewer (p. 8).

Comment 2: However, the biological assays part of the report does not seem to contribute much to the manuscript.

Response: We agree with the Reviewer that the synthesis, characterization, and DFT study of these compounds is the most important part of the manuscript and helped us reach an interesting conclusion and meet the IJMS Special Issue criteria. Therefore, we focused mainly on the effects of intramolecular hydrogen bonding on the conformational properties of ferrocene-containing peptides and emphasized this as suggested by the title of the article.

As for the biological part, much work has been done, and we are aware that the results presented are relatively modest. However, they are very similar to those reported for ferrocene peptidomimetics in our previous publications (Molecules 2014; Dalton Trans. 2015; Appl. Organometal. Chem. 2016; Appl. Organomet. Chem. 2017; Int. J. Mol. Sci. 2021; Organometallics 2022.). Although our biological results obtained in the last eight years have not yet yielded significantly active ferrocene derivatives, they are relevant because we now know which structural modifications in ferrocene peptidomimetics do not contribute to improved biological activity and are therefore not considered for further research.

From a broader perspective, we would like to emphasize that our ongoing scientific project aims to establish a library of ferrocene peptidomimetics with a clear relationship between structure and biological activity (https://croris.hr/projekti/projekt/6329?lang=en. This has already been explicitly mentioned at the end of the conclusion, p. 27, lines 952-955). Therefore, these preliminary results, although modest, are important and provide clues to further structural modifications needed to improve biological activity. For example, considering the influence of chirality and lipophilicity on biological activity, our group is working on the synthesis of heterochiral analogs of peptides 32-37 to determine the influence of the peptide backbone chirality on biological activity. We will also prepare their depsipeptide analogs to investigate whether replacing of amide groups with less polar ester groups contributes to improved activity.

To conclude, in the revised version of the manuscript, we have slightly rewritten the abstract (p. 1, lines 31-33) and conclusion (p. 27, line 948) to point out that these results are only preliminary and require further investigation.

Comment 3: The authors present 3 activity assays: antitumor, antimicrobial and antioxidant.

In the antitumor part, the authors present an evaluation of cytotoxicity. In this case, the graphs presented by the authors are striking, since the incubation time at which the viability measurements were taken does not appear, and the viability values above 100 % are very surprising. Also, the appearance of negative viability values in the assays in the MCF-7 line. In the case that these values were correct, the authors should explain the reason for these values or if it is due to the methodology used. In this case, the methodology and the associated calculation should be revised. 

Response: The time of incubation in the MTT assay is clearly indicated in the Figure legend and in the Methods section (i.e. 72 h) (p. 16, line 492; p. 24, line 820).

Although values above 100% in the viability assay seem unusual, it must be kept in mind that this is a biological experiment with live cells and both errors of technical and biological replicates are not negligible (e.g., https://www.mdpi.com/1422-0067/23/9/4777/html, https://www.bibliomed.org/mnsfulltext/19/19-1406166709.pdf?1655108722). However, one needs to primarily take a look at the survival curve as a whole. In addition, it is known that some compounds can stimulate growth at lower concentrations but inhibit it at higher concentrations, resulting in bell-shaped survival curves. However, we did not test this further for compounds 33 and 36, which showed high percentages at the lowest concentrations tested.

The MTT assay performed in this study is a modification of a test that has been performed for more than 20 years at the NIH/NCI Developmental Therapeutics Program. Values between 100 and 0 indicate an inhibitory effect, while negative data indicate a toxic effect of the compound (i.e. if you treat 1 million cells and after 72h of incubation readout is 200000 cells, then it would calculate to -80%). This is clearly explained in the Methods section, and properly referenced (Ref 84: Boyd, M.R.; Kenneth, D.P. Some practical considerations and applications of the national cancer institute in vitro anticancer drug discovery screen. Drug Dev. Res. 1995, 34, 91e109, https://doi.org/10.1002/ddr.430340203. See also Marjanović et al. J. Med. Chem. 2007, 50, 5, 1007–1018, https://doi.org/10.1021/jm061162u).

Comment 4: It appears that compounds 32 and 35 are the most active in the MCF7 line, 37 the most active in the HeLa line with the Hep62 line being insensitive to these compounds. Despite obtaining more interesting values in the MCF-7 line, the authors have continued working with the Hela cell line for the following assays.

Response: This observation is correct. In this preliminary work, we chose to examine the effect of compound 37 on apoptosis and cell cycle of HeLa cells instead of testing compounds 32 or 35 in the MCF-7 line, mainly because of the technical drawbacks. We have established protocols for HeLa cells, but the protocols for MCF-7 cells still need to be more optimised. Therefore, we plan to test compounds 32 and 35 and their heterochiral analogues in the MCF-7 line in our further study (see Manuscript, p. 18, line 556). We will also try to establish protocols to test the biological effect of these compounds on different types of cell lines.

Comment 5: The apoptosis and cell cycle assays provided do not obtain conclusive results of any kind. Neither the tested compound generates a relevant apoptosis nor a sufficient cell cycle arrest to qualify as "antitumor" to correlate with the presented cytotoxicity. Clearly, this is a preliminary assessment, but the authors should consider what these results add to their manuscript. In the case of assays performed by flow cytometry, I would recommend that the authors include the cytometry graphs with an analysis performed to justify the graphs presented.

Response: We agree with the reviewer that the apoptosis and cell cycle assays do not reach high statistical significance, but these preliminary results are relevant as guidance for further structural modifications required to obtain active compounds.

As suggested by the reviewer, we have included the cytometry graphs in the Supplementary Material (Figures S54 and S55).

Comment 6: Similarly, subsequent activity (antioxidant activity and antimicrobial) assays also contribute nothing relevant to the compounds.

Response: As mentioned above, our research project is focused on SAR studies of ferrocene peptidomimetics, and therefore all new compounds are subjected to antitumor, antioxidant and antimicrobial evaluation.

To the best of our knowledge, there are no literature data on antioxidant activity for this class of ferrocene peptides. Although the tested compounds did not show significant antioxidant and antimicrobial potential, we consider these preliminary results relevant in terms of further structural optimization to overcome the lack of biological activity. In addition, our further research will focus on finding other suitable methods to test both the antimicrobial activity and the inhibitory effect of ferrocene peptides on microbial cells.

Comment 7: Authors must consider presenting, for example, the same assays for compounds 32 and 35 in the MCF-7 line. If better results are collected for these compounds, it will be of interest to include the studies in the manuscript.

Response: At the moment, we are working on the synthesis of the six heterochiral analogues of the homochiral peptides 32-37 described here. Our plan for future studies is to test the biological effect of the homochiral and heterochiral peptides in the MCF-7 cell line. Considering the sensitivity of the MCF-7 cell line and the complexity of the cytological analyses, we will test the cytotoxicity of these stereoisomeric compounds and their effect on apoptosis and cell cycle in the MCF-7 line simultaneously using the same assays and under the same conditions to obtain consistent and reliable results. Since the time-consuming synthesis of the heterochiral analogues is still ongoing, we cannot expect that this overall study will be completed by the manuscript submission deadline.

Comment 8: I recommend a major revision of the article before acceptance of the report for publication.

Response: Dear Reviewer, we would like to thank you for your constructive comments and suggestions.

Round 2

Reviewer 2 Report

In the revised version of the manuscript "Hydrogen bonding drives helical chirality via 10-membered rings in dipeptide conjugates of ferrocene-1,1'-diamine" the authors have answered most of the questions addressed by this reviewer.

Even though no new activity studies are included, the justification provided by the authors for the inclusion of the existing assays in the initial manuscript seems reasonable and could be considered to be acceptable.

One of the issues proposed by this reviewer was the inclusion of the apoptosis and cell cycle plots. In this case, two graphs have been included in the supplementary figures. These plots have major flaws that need to be corrected for acceptance of the manuscript:

Figure S54 is an annexin/PI plot where it is not indicated which concentration was used by the assay and a graph with control cells is not included. The failure to include a control in the figure is a major flaw.

On the other hand, Figure S55 does include a control but in this case the legend is confusing: "treated by exposure to UV light". 

I encourage the authors to correct these errors to optimize the quality of their manuscript. 

Author Response

Dear Editor,

Thank you for giving us the opportunity to submit a revised draft of our manuscript entitled "Hydrogen bonding drives helical chirality via 10-membered rings in dipeptide conjugates of ferrocene-1,1’-diamine" for the special issue "Recent Advance in Hydrogen Bonding" in the International Journal of Molecular Sciences.

We are grateful to you and the reviewers for the time and effort you took to evaluate our manuscript.

We address the reviewers’ comments and concerns below.

Sincerely,

Lidija Barišić

Reviewer 2

Comment 1: In the revised version of the manuscript "Hydrogen bonding drives helical chirality via 10-membered rings in dipeptide conjugates of ferrocene-1,1'-diamine" the authors have answered most of the questions addressed by this reviewer.

Even though no new activity studies are included, the justification provided by the authors for the inclusion of the existing assays in the initial manuscript seems reasonable and could be considered to be acceptable.

One of the issues proposed by this reviewer was the inclusion of the apoptosis and cell cycle plots. In this case, two graphs have been included in the supplementary figures. These plots have major flaws that need to be corrected for acceptance of the manuscript:

Figure S54 is an annexin/PI plot where it is not indicated which concentration was used by the assay and a graph with control cells is not included. The failure to include a control in the figure is a major flaw.

On the other hand, Figure S55 does include a control but in this case the legend is confusing: "treated by exposure to UV light". 

I encourage the authors to correct these errors to optimize the quality of their manuscript. 

Response: Figures S54 and S55 are now corrected (Supplementary Material, pp. 70 and 71).

We would like to thank you again for your constructive comments and suggestions.
